# A Novel Hybrid Algorithm Based on Grey Wolf Optimizer and Fireworks Algorithm

**DOI:** 10.3390/s20072147

**Published:** 2020-04-10

**Authors:** Zhihang Yue, Sen Zhang, Wendong Xiao

**Affiliations:** 1School of Automation and Electrical Engineering, University of Science and Technology Beijing, Beijing 100083, China; zhyue0101@163.com (Z.Y.); wdxiao@ustb.edu.cn (W.X.); 2Key Laboratory of Knowledge Automation for Industrial Processes of Ministry of Education, School of Automation and Electrical Engineering, University of Science and Technology Beijing, Beijing 100083, China

**Keywords:** Grey Wolf Optimizer, Fireworks Algorithm, hybrid algorithm, exploitation and exploration

## Abstract

Grey wolf optimizer (GWO) is a meta-heuristic algorithm inspired by the hierarchy of grey wolves (Canis lupus). Fireworks algorithm (FWA) is a nature-inspired optimization method mimicking the explosion process of fireworks for optimization problems. Both of them have a strong optimal search capability. However, in some cases, GWO converges to the local optimum and FWA converges slowly. In this paper, a new hybrid algorithm (named as FWGWO) is proposed, which fuses the advantages of these two algorithms to achieve global optima effectively. The proposed algorithm combines the exploration ability of the fireworks algorithm with the exploitation ability of the grey wolf optimizer (GWO) by setting a balance coefficient. In order to test the competence of the proposed hybrid FWGWO, 16 well-known benchmark functions having a wide range of dimensions and varied complexities are used in this paper. The results of the proposed FWGWO are compared to nine other algorithms, including the standard FWA, the native GWO, enhanced grey wolf optimizer (EGWO), and augmented grey wolf optimizer (AGWO). The experimental results show that the FWGWO effectively improves the global optimal search capability and convergence speed of the GWO and FWA.

## 1. Introduction

Finding an optimal solution in high-dimensional complex space is a common issue in many engineering fields [1]. When solving such problems, deterministic algorithms find it difficult to find the optimal value, the calculation cost is too high, and the calculation time is too long [2]. The meta-heuristic optimization algorithm has, due to its simplicity and strong searching ability, been widely used in solving complex problems. In recent years, scholars around the world have done a lot of research on these algorithms. Many natural-inspired meta-heuristic algorithms have been proposed, such as Particle Swarm Optimization (PSO) [3], Ant Colony Optimization (ACO) [4], Artificial bee colony algorithm (ABC) [5], Whale Optimization Algorithm (WOA) [6], Bird Swarm Algorithm (BSA) [7], Grey Wolf Optimizer (GWO) [8], Fireworks Algorithm (FWA) [9], Biogeography-based optimization (BBO) [10], and Moth Flame Optimization (MFO) [11].

Inspired by the leadership hierarchy and the mechanism of hunting of grey wolves, Gray Wolf Optimizer, a new meta-heuristic optimization algorithm, was proposed by Mirjalili in 2014 [8]. The GWO algorithm mimics the hunting mechanism to search the optima. In [8], several benchmark functions are used to evaluate the performance of GWO. The experimental results show that the GWO algorithm is feasible and superior to PSO, Gravitational Search Algorithm (GSA) [12], and Differential Evolution (DE) [13] in the accuracy of the solution and convergence speed. Due to the advantages of fewer adjustment parameters and a faster convergence, the GWO algorithm has been applied in a series of engineering problems such as the speed control of DC motors [14], parameter estimation in surface waves [15], and load frequency control of the Multi-microgrid System [16]. However, when facing problems with multi optimal solutions like multidimensional feature selection, GWO converges to local optima and fails to find the global optimal solution. Inspired by observing fireworks explosions, Tan and Zhu proposed the fireworks algorithm (FWA) in 2010. FWA mimics the explosion of fireworks that produce sparks and illuminate the surrounding area. The global optimal solution was found by controlling the amplitude of the explosion and the number of sparks generated by explosion. Several benchmark functions are employed to test the competence of FWA. The experimental results [9] show that FWA has a better solution accuracy and faster convergence speed than SPSO (Standard particle swarm optimization) [17] and CPSO (Clonal particle swarm optimization) [18].

When searching for the optimal value in a high-dimensional space, the single meta-heuristic algorithm always has some disadvantages, such as a low accuracy, poor generalization ability, and poor local optima avoidance ability. The hybrid algorithm utilizes the differences between the two optimization algorithms, combines their advantages, and makes up for shortcomings to improve the overall performance of solving complex optimization problems [19]. The solution searched for by the genetic algorithm (GA) [20] is not accurate enough, and the evolutionary strategy (ES) tends to fall into a local minimum when searching for the optimal value. Therefore, a hybrid algorithm in [21] combined GA and ES to make up for these shortcomings. This algorithm is used for electromagnetic optimization problems and achieved satisfactory results. In [22], two hybrid models were established by Mafarja to design feature selection techniques based on WOA. In the first model, the simulated annealing (SA) algorithm is inserted into the WOA algorithm. In the second model, SA and WOA are used separately. SA is used to exploit the search space found by the WOA algorithm. In [23], Alomoush proposed a hybrid algorithm based on Gray Wolf Optimizer (GWO) and Harmony Search (HS). GWO is used to update the pitch adjustment rate and bandwidth in the HS to improve the global optimization capabilities of the hybrid algorithm. In [24], the grey wolf optimizer and crow search algorithm are combined by Sankalap Arora. The hybrid algorithm is tested on 21 data sets as a feature selection technique. The results show that the algorithm has many advantages in solving complex optimization problems. In [25], biogeography-based optimization (BBO) and differential evolution (DE) are fused by sharing population information. In BBO/DE, the migration operators in BBO are changed as the increase of the iteration count. The results indicate that the hybrid algorithm is more effective when compared to traditional BBO and DE. In [26], a novel hybrid algorithm combining grey wolf optimizer (GWO) with particle swarm optimization (PSO) is used as a load balancing technique in the cloud computing environment. The conclusions indicate that the hybrid algorithm improves the convergence speed and simplicity when compared with other algorithms. In [27], ZHU proposed a novel hybrid algorithm based on the grey wolf optimizer (GWO) and differential evolution (DE). This algorithm is tested on 23 benchmark functions and a non-deterministic polynomial hard problem. The experimental results show that this algorithm has a good performance in exploration. A global optimization algorithm combining biogeography-based optimization (BBO) with fireworks algorithm (FWA) is proposed in [28]. The BBO migration operators have been inserted into the FWA to enhance the information exchange between the population and to improve the global optimization capability. In [29], WOA-SCA, composed of the whale optimization algorithm (WOA) and sine cosine algorithm (SCA) is proposed. SCA and WOA are used for exploration and exploitation, respectively. The WOA-SCA optimization algorithm is used to distribute DGs (distributed generators) and DSTATCOMs (distribution flexible alternating current transmission devices) to enhance the voltage profile by minimizing the total power losses in the system.

As mentioned above, GWO advances itself strongly on exploitation. However, in some cases it converges prematurely and gets trapped in a local optimum. FWA not only has a high exploration capability, but also has the disadvantage of a slow convergence speed compared to the recent algorithms. This paper proposes a new hybrid algorithm which combines the exploration ability of FWA with the exploitation ability of GWO to increase the convergence characteristics. 

The major work of this paper is summarized as follows:(1)A novel hybrid algorithm based on GWO and FWA is proposed.(2)The proposed algorithm is tested on 16 benchmark functions with a wide range of dimensions and varied complexities.(3)The performance of the proposed approach is compared with standard GWO, FWA, Moth Flame Optimization (MFO), Crow Search Algorithm (CSA) [30], improved Particle Swarm Optimization(IPSO) [31], Biogeography-based optimization (BBO), Particle Swarm Optimization (PSO), Enhanced GWO (EGWO) [32], and Augmented GWO (AGWO) [33].

The rest of paper is arranged as follows: Section 2 introduces the GWO algorithm and FWA algorithm used in this paper. In Section 3, the FWGWO hybrid algorithm is proposed. Section 4 shows the experimental results and comparison of the algorithms used in the test function. Finally, the conclusions are given in Section 5.

## 2. Algorithms

### 2.1. Grey Wolf Optimizer

The Grey Wolf Optimizer (GWO) is a meta-heuristic algorithm proposed by Mirgalili et al. [8] in 2014. The GWO algorithm mimics the hunting mechanism and the leadership hierarchy of wolves to search the optima. Grey wolves are social animals, with an average of 5 to 12 wolves in each group. They also have a strict hierarchy. There are four levels in wolves’ hierarchy, called alpha (α), beta (β), delta (δ), and omega (ω). Alpha is the first level. It is responsible for making decisions like hunting, finding a place to sleep, the waking time, and so on. The second level is alpha’s candidate, the beta wolves, which help alpha in making decisions or in engaging in other activities. The third level is delta. The delta wolves are under the command of the first two levels, mainly responsible for reconnaissance, sentry, guard, and other tasks. The last level of a pack is omega. Omega wolves have to submit to the wolves in the first three levels. The omega wolves maintain the integrity of the hierarchical structure. The mathematical model of the GWO algorithm’s hierarchy and hunting behavior is as follows: 

#### 2.1.1. Hierarchical Structure

The GWO algorithm is a mathematical model based on the social hierarchy of wolves. The fittest solution found is considered as the alpha (α). The second and third best solutions are beta (β) and delta (δ), respectively. The rest of the solutions are omega (ω). In the GWO algorithm, alpha, beta, and delta collectively command omega to search for the solution space.

#### 2.1.2. Encircling Prey

The grey wolf encircles the prey when hunting. The mathematical equations for this behavior are shown in Equations (1) and (2):
(1)$$\vec{D}=\left|\vec{C}\cdot \vec{X_{P}}(t)-\vec{X}(t)\right|$$
(2)$$\vec{X}(t+1)=\vec{X_{P}}(t)-\vec{A}\cdot \vec{D}$$
where $\vec{D}$ is the distance from the wolf to the prey. $\vec{X}$ and $\vec{X_{p}}$ represent the position vector of the grey wolf and the position vector of the prey, respectively. t indicates the current iteration. $\vec{A}$ and $\vec{C}$ are coefficient vectors and are calculated as follows:(3)$$\vec{A}=2\vec{a}\cdot \vec{r_{1}}-\vec{a}$$
(4)$$\vec{C}=2\cdot \vec{r_{2}}$$
where $\vec{a}$ decreases linearly with the number of iterations from 2 to 0. $\vec{r_{1}}$ and $\vec{r_{2}}$ are the random vectors in [0,1].

#### 2.1.3. Hunting

The grey wolves can easily encircle the prey with the ability to recognize its location. The whole hunting process is usually led by the alpha. However, in a complex search space, it is impossible to get the location of the prey at the beginning. Therefore, GWO consider that the first three best solutions, alpha, beta, and delta, have more information about the location of the prey. Then, the other wolves update their positions based on these three positions, as shown in Figure 1.

The mathematical equations of this phase are as follows:(5)$$\vec{D_{\alpha }}=\left|{\vec{C}}_{1}\cdot {\vec{X}}_{\alpha }-\vec{X}\right|,\vec{D_{\beta }}=\left|{\vec{C}}_{2}\cdot {\vec{X}}_{\beta }-\vec{X}\right|,\vec{D_{\delta }}=\left|{\vec{C}}_{3}\cdot {\vec{X}}_{\delta }-\vec{X}\right|$$
(6)$${\vec{X}}_{1}={\vec{X}}_{\alpha }-{\vec{A}}_{1}\cdot ({\vec{D}}_{\alpha }),{\vec{X}}_{2}={\vec{X}}_{\beta }-{\vec{A}}_{2}\cdot ({\vec{D}}_{\beta }),{\vec{X}}_{3}={\vec{X}}_{\delta }-{\vec{A}}_{3}\cdot ({\vec{D}}_{\delta })$$
(7)$${\vec{X}}_{\left(t+1\right)}=\frac{{\vec{X}}_{1}+{\vec{X}}_{2}+{\vec{X}}_{3}}{3}$$
where, ${\vec{X}}_{\alpha }$, ${\vec{X}}_{\beta }$, and ${\vec{X}}_{\delta }$ are the position vectors of alpha, beta, and delta; $\vec{D_{\alpha }}$, $\vec{D_{\beta }}$, and $\vec{D_{\delta }}$ represent the distances from the search agent to alpha, beta, and delta respectively; ${\vec{C}}_{1}$, ${\vec{C}}_{2}$, and ${\vec{C}}_{3}$ are coefficient vectors; and ${\vec{X}}_{1}$, ${\vec{X}}_{2}$, and ${\vec{X}}_{3}$ are the step lengths in the direction toward alpha, beta, and delta. $\vec{X}$ and ${\vec{X}}_{\left(t+1\right)}$ indicate the position of the search agent before and after the update.

#### 2.1.4. Search for Prey (Exploration) and Attacking Prey (Exploitation) 

When the prey stops moving, the grey wolf completes the hunting process by attacking. In order to mimic the process of the grey wolf approaching the prey, the GWO algorithm causes $\vec{a}$ to linearly decrease from 2 to 0, as shown in Equation (8):(8)$$\vec{a}=2-(2\times t/Max_{iter})$$

According to Equation (3), $\vec{A}$ is a random value that lies in the range [−2a, 2a]. GWO uses $\vec{A}$ to force wolves to move closer or farther away from their prey. If $\vec{A}$ < 1, the wolf will be forced to attack towards the prey. If $\vec{A}$ > 1, the wolf will be forced to diverge from the prey (local minimum) to find a new fitter prey. $\vec{C}$ is a random value that lies in the range [0,2] which is employed to help GWO avoid being trapped in the local optima. The pseudo code of the GWO algorithm is presented in Algorithm 1.
Algorithm 1 Pseudo Code of GWO$\begin{array}{l} 1.\mathrm{Initialize}\;\mathrm{the}\;\mathrm{wolf}\;\mathrm{population}\;X_{i}(i=1,2,\ldots ,n)\\ 2.\mathrm{Initialize}\;a,A,\mathrm{and}\;C\\ 3.\;\mathrm{Calculate}\;\mathrm{the}\;\mathrm{fitness}\;\mathrm{of}\;\mathrm{each}\;\mathrm{search}\;\mathrm{agent}\\ 4.\;X_{\alpha }=\mathrm{the}\;\mathrm{best}\;\mathrm{search}\;\mathrm{agent}\\ 5.\;X_{\beta }=\mathrm{the}\;\sec\mathrm{ond}\;\mathrm{best}\;\mathrm{search}\;\mathrm{agent}\\ 6.\;X_{\delta }=\mathrm{the}\;\mathrm{third}\;\mathrm{best}\;\mathrm{search}\;\mathrm{agent}\\ 7.\;\mathrm{while}\;(t<\mathrm{Max}\;\mathrm{number}\;\mathrm{of}\;\mathrm{iterations})\\ 8.\;\mathrm{for}\;\mathrm{each}\;\mathrm{search}\;\mathrm{agent}\\ 9.\;\mathrm{Update}\;\mathrm{the}\;\mathrm{position}\;\mathrm{of}\;\mathrm{the}\;\mathrm{current}\;\mathrm{search}\;\mathrm{agent}\;\mathrm{by}\;\mathrm{equation}(7)\\ 10.\;\mathrm{end}\;\mathrm{for}\\ 11.\;\mathrm{Update}a,A,\mathrm{and}C\\ 12.\;\mathrm{Calculate}\;\mathrm{the}\;\mathrm{fitness}\;\mathrm{of}\;\mathrm{all}\;\mathrm{search}\;\mathrm{agents}\\ 13.\;\mathrm{Update}X_{\alpha },X_{\beta },\mathrm{and}\;X_{\delta }\\ 14.\;t=t+1\\ 15.\;\mathrm{end}\;\mathrm{while}\\ 16.\;\mathrm{return}\;X_{\alpha } \end{array}$


## 2.2. Fireworks Algorithm

Inspired by the behavior according to which fireworks burst into sparks in the night sky and illuminate the surrounding area, Tan and Zhu proposed the Fireworks Algorithm (FWA) in 2010. In FWA, a firework is considered as a viable solution in the search space of the optimization problem. Furthermore, the process of sparking fireworks is deemed as the process of searching the neighborhood of these fireworks. The specific steps of the fireworks algorithm are as follows:(1)N fireworks are randomly generated in the search space, and each firework represents a feasible solution.(2)Evaluate the quality of these fireworks. Consider that the optimal location may be close to the fireworks with a better fitness; these fireworks get a smaller search amplitude and more explosion sparks to search the surrounding area. On the contrary, those fireworks with a bad fitness will get fewer explosion sparks and a larger search amplitude. The number of explosion sparks and the explosion amplitude of fireworks are calculated as shown in Equations (9) and (10):
(9)$$S_{i}=m\cdot \frac{y_{\max}-f(x_{i})+\xi }{\sum_{i=1}^{n}(y_{\max}-f(x_{i}))+\xi }$$
(10)$$A_{i}=\hat{A}\cdot \frac{f(x_{i})-y_{\min}+\xi }{\sum_{i=1}^{n}(f(x_{i})-y_{\min})+\xi }$$
where $S_{i}$ is the number of explosion sparks. $A_{i}$ represents the amplitude of the explosion. $y_{\min}=\min(f(x_{i})),(i=1,2,\ldots ,N)$ is the minimum fitness among the N fireworks. $y_{\max}=\max(f(x_{i})),(i=1,2,\ldots ,N)$ is the maximum fitness value among the N fireworks. m and $\hat{A}$ are parameters controlling the total number of sparks and the maximum explosion amplitude, respectively. ξ is the smallest constant in the computer, utilized to avoid a zero-division-error. To avoid that the good fireworks produce far more explosive sparks than the fireworks with a poor fitness, Equation (11) is used to bound the number of sparks that are generated:(11)$${\hat{S}}_{i}=\left\{ \begin{array}{ll} round(a\cdot m) & \;{\mathrm{if}\;S}_{i}<am\\ round(b\cdot m) & \;{\mathrm{if}\;S}_{i}>bm,a<b<1\\ round(S_{i}) & \;{\mathrm{if}\;S}_{i}\;\mathrm{otherwise} \end{array}\right.$$
where a,b are constant parameters, and round() represents the rounding function.(3)To guarantee the diversity of the fireworks, another method of generating sparks, Gaussian explosion, is designed in FWA. For the randomly selected fireworks, a number of dimensions are randomly selected and updated, as shown in Equation (12), to get the position of a Gaussian spark at the dimension *k*:(12)$${\hat{x}}_{ik}=x_{ik}\times e$$
where e∼N(1,1), N(1,1) is a Gaussian random value with mean 1 and standard deviation 1.The generated explosion sparks and Gaussian sparks may exceed the boundaries of the search space. Equation (13) maps the sparks beyond the boundary at dimension *k* to a new position:(13)$${\tilde{x}}_{k}^{j}=x_{k}^{\min}+\left|{\tilde{x}}_{k}^{j}\right|\%(x_{k}^{\max}-x_{k}^{\min})$$
where $x_{k}^{\max}$ represents the upper bound of the search space at dimension *k*, and $x_{k}^{\min}$ is the lower bound of the search space at dimension *k*.(4)N locations should be selected as the next generation of fireworks from the explosion sparks, Gaussian sparks, and the current fireworks. In FWA, the location with the best fitness is always kept for the next iteration. Then, N−1 locations are chosen, determined by their distance to other locations. The distance and the selection probability of $x_{i}$ is defined as follows:(14)$$R(x_{i})=\sum_{j\in k}d(x_{i},x_{j})=\sum_{j\in k}\left\|x_{i}-x_{j}\right\|$$
(15)$$p(x_{i})=\frac{R(x_{i})}{\sum_{j\in k}R(x_{j})}$$

In this selection strategy, if there are many other locations around $x_{i}$, the selection probability will be reduced to keep the diversity of the next generation.

The execution flow of the FWA algorithm is shown in Algorithm 2.
Algorithm 2 Pseudo Code of FWA$\begin{array}{l} 1.\;\mathrm{Randomly}\;\mathrm{select}\;n\;\mathrm{locations}\;\mathrm{for}\;\mathrm{fireworks}\\ 2.\;\mathrm{while}\;\mathrm{stop}\;\mathrm{criteria}=\mathrm{flase}\;\mathrm{do}\\ 3.\;\mathrm{Set}\;\mathrm{off}\;n\;\mathrm{fireworks}\;\mathrm{respectively}\;\mathrm{at}\;\mathrm{the}\;n\;\mathrm{locations}\\ 4.\;\mathrm{for}\mathrm{each}\;\mathrm{firework}x_{i}\mathrm{do}\\ 5.\;\mathrm{Calculate}\;\mathrm{the}\;\mathrm{number}\;\mathrm{of}\;\mathrm{sparks}\;\mathrm{that}\;\mathrm{the}\;\mathrm{firework}\;\mathrm{yields}\;\mathrm{by}\;\mathrm{equation}\;(9)\\ 6.\;\mathrm{Obtain}\;\mathrm{locations}\;\mathrm{of}\;\mathrm{each}\;\mathrm{sparks}\;\mathrm{of}\;\mathrm{the}\;\mathrm{firework}x_{i}\\ 7.\;\mathrm{end}\;\mathrm{for}\\ 8.\;\mathrm{for}\;k=1:\mathrm{number}\;\mathrm{of}\;\mathrm{Gaussian}\;\mathrm{sparks}\;\mathrm{do}\\ 9.\;\mathrm{Randomly}\;\mathrm{select}\;a\;\mathrm{firework}x_{j}\\ 10.\;\mathrm{Generate}\;a\;\mathrm{Gaussian}\;\mathrm{spark}\;\mathrm{for}\;\mathrm{the}\;\mathrm{firework}\\ 11.\;\mathrm{end}\;\mathrm{for}\\ 12.\;\mathrm{Select}\;\mathrm{the}\;\mathrm{best}\;\mathrm{location}\;\mathrm{and}\;\mathrm{keep}\;\mathrm{it}\;\mathrm{for}\;\mathrm{next}\;\mathrm{explosion}\;\mathrm{generation}\\ 13.\;\mathrm{Randomly}\;\mathrm{select}\;n-1\;\mathrm{locations}\;\mathrm{from}\;\mathrm{the}\;\mathrm{two}\;\mathrm{types}\;\mathrm{of}\;\mathrm{sparks}\;\mathrm{and}\;\mathrm{the}\;\mathrm{current}\;\mathrm{fireworks}\\ 14.\;\mathrm{according}\;\mathrm{to}\;\mathrm{the}\;\mathrm{probability}\;\mathrm{given}\;\mathrm{in}\;\mathrm{equation}\;(15)\\ 15.\;\mathrm{end}\;\mathrm{while} \end{array}$

## 3. Hybrid FWGWO Algorithm

### 3.1. Establishment of FWGWO

As mentioned in Section 2, GWO is strong at exploitation but weak at avoiding a premature convergence and local optimum. The FWA algorithm has a strong exploration capability, but it lacks in exploitation. In this section, the FWGWO hybrid algorithm is proposed to combine the GWO exploitation capability with the FWA exploration capability to obtain a better global optimization capability. FWGWO alternately uses the FWA algorithm for exploration in the search space and the GWO algorithm for exploitation to search the global optimum without changing the general operation of the GWO and FWA algorithms. In order to balance the exploration with the exploitation, an adaptive equilibrium coefficient is proposed in this paper. Updating the position $X_{\alpha }$ means that the best fitness has been changed. When the current position is closer to the optimal solution, the coefficient p will be updated to change the search strategy, as shown in Equation (16):(16)$$p=0.9\times (1-\cos(\frac{\pi }{2}\cdot \frac{t}{Max_{iter}}))$$
where p represents the adaptive balance coefficient. t is the current number of iterations. $Max_{iter}$ indicates the maximum number of iterations.

A random value r in [0, 1] is set for a comparison with the adaptive balance coefficient p. If r>p, the next iteration will be executed using the FWA algorithm. Otherwise, the GWO algorithm is used for this iteration. The function curve of p is shown in Figure 2. The value of p is small and slowly increases in the early optimization stage to make sure that FWGWO explores a huge search space by multiple calls of FWA to avoid being trapped in the local minimum. In the later optimization phases, the algorithm exploits small regions to efficiently search the optimum with the rapidly increasing p. To avoid that only GWO is executed in the final stage of the FWGWO algorithm, the value of p is growing in [0, 0.9]. In some cases, after one iteration of the FWA algorithm, the FWGWO algorithm will proceed to the next iteration of the FWA algorithm without further exploitation so as to escape the current local optimal space and miss the global optimal solution. To avoid these cases, the FWGWO algorithm exploits the current region with at least T iterations of the GWO algorithm before proceeding to the next FWA algorithm. In this paper, T is set to 10. The variable k is defined to count how many GWO iterations have occurred since the last FWA iteration. k is initialized at the beginning of FWGWO. After each GWO iteration, k increases itself by 1. If k>T and r>p, FWA will be used to execute the iteration. At the last of these iterations, k will be set to 0. 

The pseudo code of the hybrid FWGWO algorithm is shown in Algorithm 3.Algorithm 3 Pseudo Code of FWGWO$\begin{array}{l} 1.\;\mathrm{Initialize}\;\mathrm{the}\;\mathrm{wolf}\;\mathrm{population}\;X_{i}(i=1,2,\ldots ,n)\\ 2.\;\mathrm{Initialize}\;a,\;A,\;k,\;t\;\mathrm{and}\;C\\ 3.\;\mathrm{Calculate}\;\mathrm{the}\;\mathrm{fitness}\;\mathrm{of}\;\mathrm{each}\;\mathrm{search}\;\mathrm{agent}\\ 4.\;X_{\alpha }=\mathrm{the}\;\mathrm{best}\;\mathrm{search}\;\mathrm{agent}\\ 5.\;X_{\beta }=\mathrm{the}\;\sec\mathrm{ond}\;\mathrm{best}\;\mathrm{search}\;\mathrm{agent}\\ 6.\;X_{\delta }=\mathrm{the}\;\mathrm{third}\;\mathrm{best}\;\mathrm{search}\;\mathrm{agent}\\ 7.\;\mathrm{while}\;(t<\mathrm{Max}\;\mathrm{number}\;\mathrm{of}\;\mathrm{iterations})\\ 8.\;\mathrm{for}\mathrm{each}\;\mathrm{search}\;\mathrm{agent}\\ 9.\;\mathrm{Update}\;\mathrm{the}\;\mathrm{position}\;\mathrm{of}\;\mathrm{the}\;\mathrm{current}\;\mathrm{search}\;\mathrm{agent}\;\mathrm{by}\;\mathrm{equation}\;(7)\\ 10.\;\mathrm{end}\;\mathrm{for}\\ 11.\;\mathrm{Update}a,A,\mathrm{and}C\\ 12.\;\mathrm{Calculate}\;\mathrm{the}\;\mathrm{fitness}\;\mathrm{of}\;\mathrm{all}\;\mathrm{search}\;\mathrm{agents}\\ 13.\;\mathrm{Update}X_{\alpha },X_{\beta },\mathrm{and}\;X_{\delta }\\ 14.\;\mathrm{if}X_{\alpha }\;\mathrm{changed}\;\mathrm{then}\\ 15.\;\mathrm{Update}\;\mathrm{the}\;\mathrm{adaptive}\;\mathrm{balance}\;\mathrm{coefficient}\;(p)\;\mathrm{by}\;\mathrm{equation}\;(16)\\ 16.\;\mathrm{end}\;\mathrm{if}\\ 17.\;\mathrm{if}\;k\;>=\;T\;\mathrm{and}\;rand()\;>\;p\;\mathrm{then}\\ 18.\;\mathrm{for}\;eachsearchagent\;x_{i}\;\mathrm{do}\\ 19.\;\mathrm{Generate}\;\mathrm{explosion}\;\mathrm{sparks}\\ 20.\;\mathrm{end}\;\mathrm{for}\\ 21.\;\mathrm{Generate}m\mathrm{Gaussian}\;\mathrm{spark}\;\mathrm{for}\;\mathrm{the}\;\mathrm{search}\;\mathrm{agents}\\ 22.\;\mathrm{Selec}\mathrm{new}\mathrm{search}\mathrm{agents}\mathrm{by}\mathrm{equation}\;(15)\\ 23.\;t=t+1\\ 24.\;k=0\\ 25.\;\mathrm{end}\;\mathrm{if}\\ 26.\;k=k+1\;t=t+1\\ 27.\mathrm{end}\;\mathrm{while}\\ 28.\mathrm{return}\;X_{\alpha } \end{array}$

### 3.2. Time Complexity Analysis of FWGWO

The time complexity of FWGWO is mainly determined by the population size n, dimensions of the solution space d, and max number of iterations t. O (FWGWO) = O (population initialization) + O (Calculation of fitness for the entire population) + O (Update of population position). The time complexity of the population initialization is O (n×d), and the time complexity for calculating the fitness of the entire population is O (t×n×d). As for the part with the updating position, the time complexity consists of two parts: O = O (position update with GWO) + O (position update with FWA). The time complexity of this part is O (10/11×t×n×d+1/11×t×3×n×d) ≈ O(t×n×d). Therefore, the time complexity of the FWGWO algorithm is O (n×d+t×n×d+t×n×d) = O ((2t+1)×n×d). For comparison, the time complexity of GWO is O ((2t+1)×n×d), and the time complexity of FWA is O ((4t+1)×n×d). The time complexity of FWGWO is the same as that of GWO and smaller than that of FWA.

## 4. Experimental Section and Results

In this section, 16 benchmark functions with different characteristics are used to evaluate the proposed FWGWO hybrid algorithm. The experiment results are compared with nine other algorithms to verify the superiority of FWGWO.

### 4.1. Compared Algorithms

A total of nine algorithms, including IPSO, PSO, BBO, CSA, MFO, FWA, GWO, AGWO, and EGWO, were selected for comparison with the FWGWO algorithm proposed in this paper. These nine algorithms contain both classical algorithms and new algorithms proposed in recent years. The parameter settings of these algorithms are shown in Table 1. These parameters are chosen based on the parameters of these algorithms in the original papers. For all algorithms in this experiment, the size of the population is 20, the dimension is 100 dimensions, and the maximum number of iterations is 500. This experiment has been carried out with different experimental parameters. The size of the population has been set to 10, 20, 30, and 50. The dimension has been set to 20, 50, 100, and 500. The results of these experiments are similar. Considering the length of this paper, the typical parameters mentioned above are taken as an example.

### 4.2. Benchmark Functions

Sixteen benchmark functions listed in Table A1 (see in Appendix A) are utilized to evaluate the optimized performance of FWGWO. These benchmark functions can be divided into two categories. In the first category, the unimodal functions f1-f8 with only one global optimum are used to test the exploitation capability. In the second category, the multimodal functions f9–f16 with more than two local optima are utilized to assess the ability of FWGWO to find global optima.

### 4.3. Performance Metrics

Three performance indicators, the mean fitness, fitness variance, and best fitness, are selected in this paper to evaluate the results of the experiments. The mathematical equations are defined as follows:(17)$$MeanFitness=\frac{1}{M}\sum_{i=1}^{M}G_{i}$$
(18)$$std=\sqrt{\frac{\sum_{i=1}^{M}{\left(G_{i}-MeanFitness\right)}^{2}}{M}}$$
where M represents the total number of independently repeated experiments. M is 30 in this paper. G represents the function fitness of each experiment, and i is the count of repeated experiments.

In addition, a nonparametric statistical test, Wilcoxon’s rank-sum test [34], is utilized to show that the proposed FWGWO algorithm provides a significant improvement over other algorithms. The test was carried out with the results of the FWGWO and other algorithms in each benchmark function at a 5% significance level. All experiments are carried out using MATLAB R2017a on a computer with Inter i5-5200U 2.2GHz and an 8 GB memory.

### 4.4. Comparison and Analysis of Simulation Results

Thirty independently repeated experiments were implemented using the 10 algorithms mentioned above on each benchmark function. Table 2 shows the test results of these algorithms on each benchmark function. The best, mean, and variance of the fitness obtained by each algorithm are listed in Table 2, and the best results are bolded. Compared with other algorithms, the average of the fitness obtained by the FWGWO algorithm after 500 iterations is better and closer to the global optimal value on most of the benchmark functions. In terms of the best fitness obtained in 30 repetitions, FWGWO has a better performance than other algorithms in all 16 functions. What is more, the best fitness obtained by FWGWO is 0 on functions f9, f11, and f16. This shows that the FWGWO algorithm has found the global optimal value on these functions. These cases verified the superiority of FWGWO over other algorithms.

Furthermore, the variances of the results of these 30 independent experiments are also smaller. As can be seen in Table 2, the optimization results of FWGWO on the unimodal functions f1–f5, whether the mean or the variance, are superior to other algorithms. As mentioned above, the unimodal function is utilized to test the exploitation ability of the algorithm. From these results, it can be seen that, compared with FWA, GWO, and other algorithms, FWGWO can find a better solution in a limited number of iterations and has a better exploitation ability. It can also be seen from Table 2 that FWGWO has better results in seven of the eight functions on the multimodal functions f9–f16. The multimodal functions are used to test the exploration ability of the algorithm. It can be known from the experiment results that the FWGWO algorithm has a better global optimization ability than other algorithms used in this paper, including the original GWO, the standard FWA, and the enhanced GWO. 

Figure 3 and Figure 4 show the convergence process of FWGWO and other algorithms mentioned in this paper on 16 reference benchmark functions. The curves we provided in Figure 3 and Figure 4 are the average fitness curves of 30 repetitions. As shown in Figure 3, at the first few iterations, FWGWO’s convergence is slightly slower than that of other algorithms. In this process, the algorithm explores a lot to search for a better exploitation of the local areas. In the following convergence process, FWGWO converges quickly in this region, searching for a better solution than other algorithms after a total of 500 iterations. Figure 4 shows the comparison of the convergence of FWGWO with other algorithms on multimodal functions. As mentioned above, the multimodal functions are used to test the local optima avoidance ability of the algorithm.

In Figure 4, we can see that unlike other algorithms that have been trapped in the local optimum, FWGWO continues to search for a better fitness on most benchmark functions and keeps approaching global optima. This proves that the FWGWO algorithm has a better global optimization searching ability when compared with other algorithms. According to the above comprehensive description, the hybrid algorithm FWGWO is superior to other algorithms in both the accuracy of the solution and the convergence. It can be concluded that the FWGWO algorithm has better overall performances on most optimization problems than other algorithms.

The mean running time of 30 repetitions has been shown in Table 3. The results show that the running time of the FWGWO algorithm is a little bigger than that of GWO and much smaller than that of FWA. As mentioned in Section 3, FWA has a larger time complexity, while GWO has a smaller time complexity. The location update strategy of FWA has been used in FWGWO a few times. Therefore, FWGWO has a bigger running time than GWO. Considering the improvement of the convergence performance, it is acceptable to sacrifice a little running time.

Wilcoxon’s nonparametric statistical test is conducted at the 5% significance level in order to determine whether the FWGWO provides a significant improvement compared to other algorithms or not. The results of different algorithms on the benchmark function were employed to test the Wilcoxon rank sum, and p and h values were obtained as significant level indicators. If the p value is less than 0.05, the null hypothesis is rejected. At this time, the h value is 1, and the two algorithms tested are considered significantly different. Conversely, when the p value is greater than 0.05, the h value is 0, and the two algorithms tested are considered to not be significantly different. In this paper, the Wilcoxon rank sum is tested with the results of 30 repeated experiments on 16 benchmark functions by the FWGWO algorithm and other algorithms. The test results are shown in Table 4. In most cases, the h values of the test results are 1, except that the results’ p values for AGWO and FWGWO on f8, f11, f15, and f16 are greater than 0.05 and the h values are 0. This means that the optimization efficiency of FWGWO and AGWO is similar in f8, f11, f15, and f16. The results show that in most cases the performance of the FWGWO algorithm is significantly improved when compared with other algorithms.

## 5. Conclusions

In this paper, a hybrid algorithm FWGWO based on Grey Wolf Optimizer (GWO) and Fireworks Algorithm (FWA) is proposed for the optimization of multidimensional complex space. The FWGWO algorithm combines the good exploitation ability of GWO with the strong exploration ability of the FWA algorithm. In order to balance the exploitation with the exploration, an adaptive balance coefficient is employed in this algorithm. The probability of exploitation or exploration is controlled by the balance coefficient. By changing the balance coefficient, the FWGWO algorithm can avoid the local optimal value as much as possible and has a fast convergence speed. In order to verify the performance superiority of the FWGWO algorithm, the FWGWO algorithm was tested 30 times with IPSO, PSO, BBO, CSA, MFO, FWA, GWO, AGWO, and EGWO algorithms on 16 benchmark functions, and their test results were compared with each other. The compared results show that the FWGWO algorithm has a better global optimization ability and faster convergence speed compared with other algorithms. In addition, the Wilcoxon rank sum test was used to test the optimization results. The test results show that the FWGWO algorithm has a significant improvement compared to other algorithms.

This paper presents a new method for combining two algorithms. In the future, this method could be used in combination with other enhanced algorithms. Furthermore, the FWGWO can be applied to solve single- and multi-objective optimization problems like the feature selection problem.

## Figures and Tables

**Figure 1 sensors-20-02147-f001:**
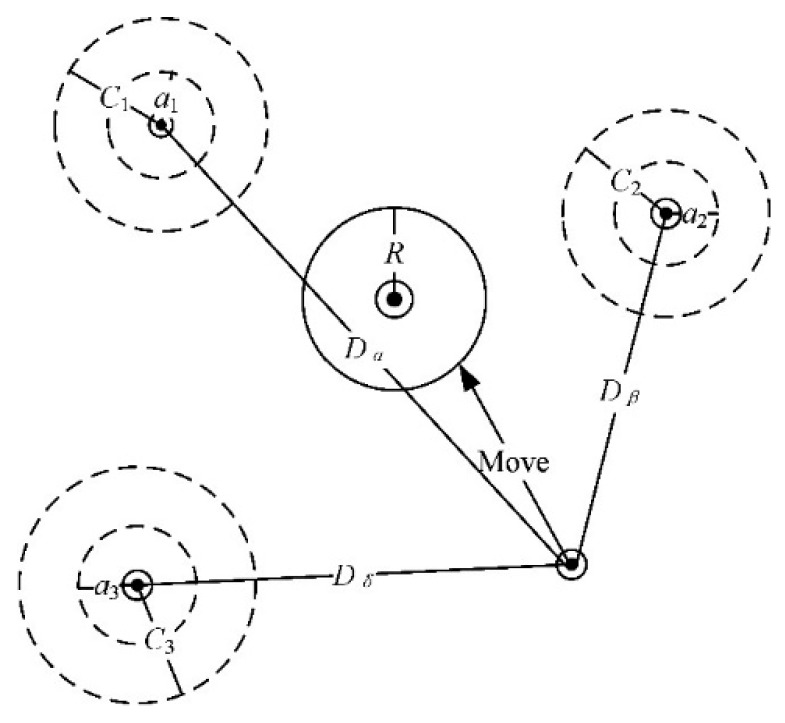
Position updating in GWO.

**Figure 2 sensors-20-02147-f002:**
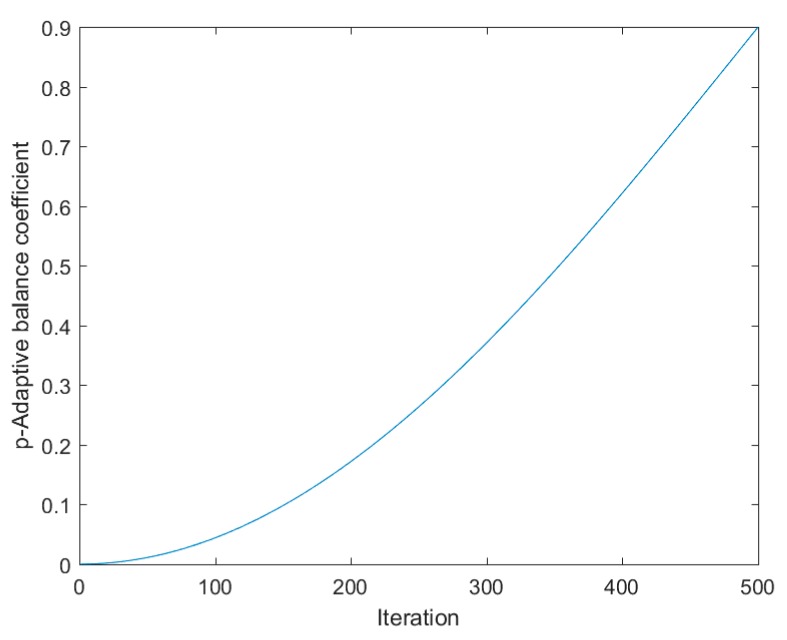
Adaptive balance coefficient.

**Figure 3 sensors-20-02147-f003:**
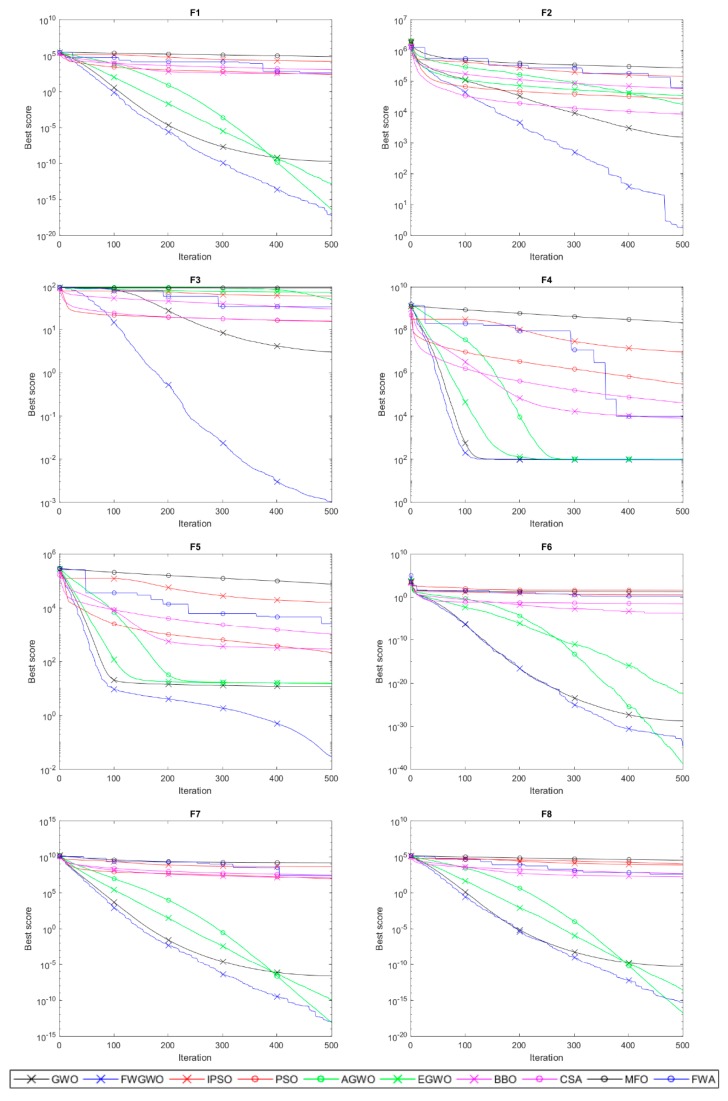
Convergence curves of the unimodal functions.

**Figure 4 sensors-20-02147-f004:**
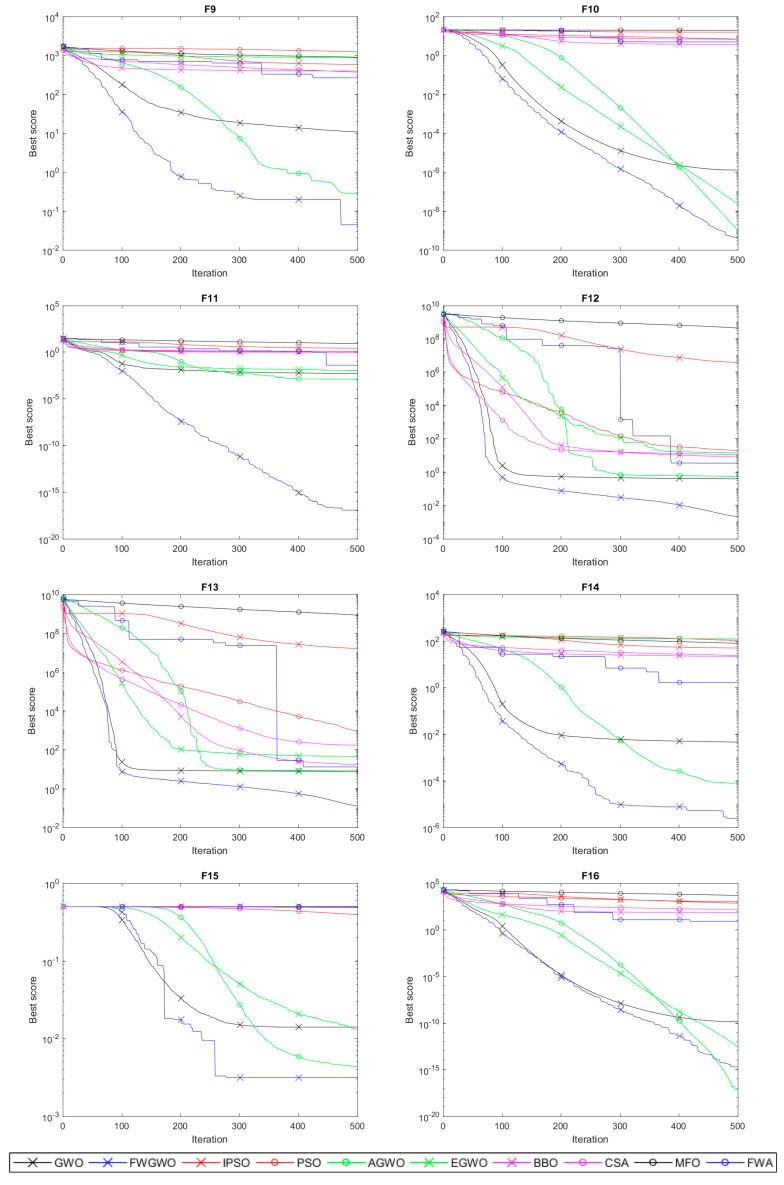
Convergence curves of the multimodal functions.

**Table 1 sensors-20-02147-t001:** Parameter settings.

	Parameter	Value(s)
GWO	a	Linearly decreased from 2 to 0
FWA	Total number of sparks	50
	Maximum explosion amplitude	40
	Number of Gaussian sparks	5
	a	0.04
	b	0.8
ISPO	Inertia w(wMin,wMax)	[0.4,0.9]
	Acceleration constants(c1,c2)	[2,2]
PSO	Inertia w(wMin,wMax)	[0.6,0.9]
	Acceleration constants(c1,c2)	[2,2]
AGWO	a	a=2−(cos(rand())×t/Max_iter)
EGWO	a	a=rand()
BBO	Immigration probability	[0,1]
	Mutation probability	0.005
	Habitat modification probability	1.0
	Step size	1.0
	Migration rate	1.0
	Maximum immigration	1.0
CSA	Flight length	2
	Awareness probability	0.1
MFO	a	a=−1+Iteration×((−1)/Max_iteration)

**Table 2 sensors-20-02147-t002:** Results of the benchmark functions.

Function		GWO	FWA	IPSO	PSO	AGWO	EGWO	BBO	CSA	MFO	FWGWO
**F1**	Mean	1.99 × 10^−10^	3.73 × 10^2^	1.57 × 10^4^	2.21 × 10^2^	3.93 × 10^−17^	1.45 × 10^−13^	2.87 × 10^2^	1.06 × 10^3^	7.04 × 10^4^	**5.16 × 10^−18^**
	Std	1.19 × 10^−10^	5.49 × 10^3^	5.18 × 10^3^	3.01 × 10^1^	7.20 × 10^−17^	6.23 × 10^−14^	2.47 × 10^1^	1.83 × 10^2^	1.35 × 10^4^	1.93 × 10^−16^
	Best	2.70 × 10^−11^	4.84	7.40 × 10^3^	1.50 × 10^2^	4.61 × 10^−18^	6.15 × 10^−16^	2.54 × 10^2^	7.09 × 10^2^	4.69 × 10^4^	**6.68 × 10^−24^**
F2	Mean	1.53 × 10^3^	6.14 × 10^4^	1.44 × 10^5^	2.71 × 10^4^	1.79 × 10^4^	3.34 × 10^4^	5.74 × 10^4^	8.49 × 10^3^	2.70 × 10^5^	**1.74**
	Std	1.35 × 10^3^	7.11 × 10^4^	3.11 × 10^4^	6.72 × 10^3^	1.81 × 10^4^	1.41 × 10^4^	8.78 × 10^3^	1.52 × 10^3^	5.48 × 10^4^	6.01 × 10^1^
	Best	1.20 × 10^2^	1.32 × 10^1^	8.33 × 10^4^	1.55 × 10^4^	2.54 × 10^2^	6.75 × 10^3^	4.38 × 10^4^	5.54 × 10^3^	1.64 × 10^5^	**4.97 × 10^−7^**
F3	Mean	3.03	3.36 × 10^1^	6.00 × 10^1^	1.54 × 10^1^	4.99 × 10^1^	7.28 × 10^1^	2.99 × 10^1^	1.57 × 10^1^	9.41 × 10^1^	**1.02 × 10^−3^**
	Std	2.17	1.44 × 10^1^	4.32	1.66	3.13 × 10^1^	7.82	3.01	1.40	1.72	3.06 × 10^−3^
	Best	2.24 × 10^−1^	8.24 × 10^−1^	4.84 × 10^1^	1.26 × 10^1^	6.56 × 10^−1^	5.61 × 10^1^	2.35 × 10^1^	1.32 × 10^1^	9.13 × 10^1^	**5.61 × 10^−7^**
F4	Mean	9.81 × 10^1^	9.38 × 10^3^	9.12 × 10^6^	2.95 × 10^5^	9.82 × 10^1^	9.83 × 10^1^	7.86 × 10^3^	4.01 × 10^4^	2.04 × 10^8^	**9.00 × 10^1^**
	Std	5.22 × 10^−1^	2.50 × 10^6^	3.99 × 10^6^	6.54 × 10^4^	5.50 × 10^−1^	6.68 × 10^−1^	1.39 × 10^3^	1.80 × 10^4^	8.37 × 10^7^	1.74 × 10^1^
	Best	9.65 × 10^1^	1.00 × 10^2^	4.79 × 10^6^	1.87 × 10^5^	9.71 × 10^1^	9.63 × 10^1^	5.16 × 10^3^	2.26 × 10^4^	6.98 × 10^7^	**1.66 × 10^1^**
F5	Mean	1.16 × 10^1^	2.61 × 10^3^	1.59 × 10^4^	2.13 × 10^2^	1.50 × 10^1^	1.56 × 10^1^	2.86 × 10^2^	1.03 × 10^3^	7.46 × 10^4^	**3.02 × 10^−2^**
	Std	9.88 × 10^−1^	2.14 × 10^3^	4.71 × 10^3^	3.34 × 10^1^	6.26 × 10^−1^	9.66 × 10^−1^	2.88 × 10^1^	1.07 × 10^2^	1.39 × 10^4^	2.42 × 10^−2^
	Best	9.54	2.44 × 10^1^	8.20 × 10^3^	1.56 × 10^2^	1.40 × 10^1^	1.40 × 10^1^	2.26 × 10^2^	7.95 × 10^2^	5.23 × 10^4^	**1.27 × 10^−2^**
F6	Mean	1.69 × 10^−29^	1.73	3.58	3.72 × 10^1^	**2.42 × 10^−39^**	3.44 × 10^−23^	1.58 × 10^−4^	2.79 × 10^−2^	1.80 × 10^1^	2.09 × 10^−35^
	Std	2.50 × 10^−29^	4.11 × 10^−1^	2.52	1.11 × 10^1^	1.03 × 10^−35^	6.04 × 10^−22^	2.03 × 10^−3^	1.65 × 10^−2^	5.18	1.21 × 10^−34^
	Best	4.86 × 10^−34^	8.30 × 10^−8^	1.05 × 10^−1^	1.18 × 10^1^	5.32 × 10^−46^	1.38 × 10^−35^	6.48 × 10^−7^	8.35 × 10^−3^	2.00	**1.80 × 10^−46^**
F7	Mean	2.47 × 10^−7^	2.39 × 10^7^	4.13 × 10^8^	8.83 × 10^6^	**7.99 × 10^−14^**	1.13 × 10^−10^	1.14 × 10^7^	2.89 × 10^7^	1.46 × 10^9^	1.06 × 10^−13^
	Std	1.62 × 10^−7^	8.81 × 10^7^	2.66 × 10^8^	2.57 × 10^6^	8.94 × 10^−14^	1.44 × 10^−10^	2.50 × 10^6^	9.00 × 10^6^	8.26 × 10^8^	2.12 × 10^−12^
	Best	4.80 × 10^−8^	1.31 × 10^1^	6.05 × 10^7^	4.00 × 10^6^	3.74 × 10^−15^	1.15 × 10^−12^	7.24 × 10^6^	1.75 × 10^7^	1.16 × 10^8^	**1.00 × 10^−18^**
F8	Mean	5.38 × 10^−11^	3.72 × 10^2^	7.33 × 10^3^	1.02 × 10^4^	**1.80 × 10^−17^**	2.54 × 10^−14^	1.68 × 10^2^	4.68 × 10^2^	3.30 × 10^4^	4.94 × 10^−16^
	Std	4.61 × 10^−11^	7.11 × 10^2^	2.62 × 10^3^	1.30 × 10^3^	2.04 × 10^−17^	1.31 × 10^−14^	2.58 × 10^1^	7.19 × 10^1^	7.03 × 10^3^	5.01 × 10^−16^
	Best	1.19 × 10^−11^	4.81 × 10^−2^	3.38 × 10^3^	7.58 × 10^3^	5.21 × 10^−19^	2.86 × 10^−16^	1.35 × 10^2^	3.14 × 10^2^	1.40 × 10^4^	**1.42 × 10^−21^**
F9	Mean	1.09 × 10^1^	2.66 × 10^2^	5.88 × 10^2^	1.24 × 10^3^	2.54 × 10^−1^	8.72 × 10^2^	3.97 × 10^2^	3.72 × 10^2^	8.99 × 10^2^	**4.43 × 10^−2^**
	Std	8.83	1.04 × 10^2^	6.85 × 10^1^	9.61 × 10^1^	6.46 × 10^−5^	1.80 × 10^2^	5.40 × 10^1^	4.50 × 10^1^	7.18 × 10^1^	8.31 × 10^−2^
	Best	5.96 × 10^−7^	3.06 × 10^−2^	4.79 × 10^2^	1.05 × 10^3^	**0.00**	4.45 × 10^2^	3.17 × 10^2^	3.04 × 10^2^	6.92 × 10^2^	**0.00**
F10	Mean	1.27 × 10^−6^	4.98	1.58 × 10^1^	6.96	1.06 × 10^−9^	2.37 × 10^−8^	3.71	6.80	1.99 × 10^1^	**4.21 × 10^−10^**
	Std	3.85 × 10^−7^	2.93	9.91 × 10^−1^	3.72 × 10^−1^	7.18 × 10^−10^	2.69 × 10^−8^	1.18 × 10^−1^	5.39 × 10^−1^	1.52 × 10^−1^	5.61 × 10^−10^
	Best	6.16 × 10^−7^	1.05 × 10^−2^	1.43 × 10^1^	6.06	1.97 × 10^−10^	1.55 × 10^−9^	3.49	6.07	1.95 × 10^1^	**2.93 × 10^−14^**
F11	Mean	4.78 × 10^−3^	3.71 × 10^−2^	2.52	1.03	1.20 × 10^−3^	1.00 × 10^−2^	8.33 × 10^−1^	1.09	7.73	**1.11 × 10^−17^**
	Std	1.09 × 10^−2^	4.83 × 10^−1^	3.68 × 10^−1^	2.44 × 10^−2^	9.13 × 10^−3^	1.22 × 10^−2^	3.47 × 10^−2^	1.60 × 10^−2^	1.52	1.20 × 10^−16^
	Best	1.52 × 10^−13^	3.47 × 10^−2^	1.70	9.72 × 10^−1^	**0.00**	**0.00**	7.60 × 10^−1^	1.07	4.44	**0.00**
F12	Mean	3.83 × 10^−1^	3.34	3.57 × 10^6^	1.87 × 10^1^	5.37 × 10^−1^	1.38 × 10^1^	7.95	1.10 × 10^1^	4.31 × 10^8^	**2.13 × 10^−3^**
	Std	6.78 × 10^−2^	5.95 × 10^5^	1.56 × 10^6^	4.44	4.11 × 10^−2^	8.57	2.37	3.03	1.65 × 10^8^	6.89 × 10^−3^
	Best	2.41 × 10^−1^	1.20	1.56 × 10^5^	7.79	4.06 × 10^−1^	6.15 × 10^−1^	2.33	6.06	1.34 × 10^8^	**6.35 × 10^−4^**
F13	Mean	7.37	1.21 × 10^1^	1.69 × 10^7^	8.76 × 10^2^	8.32	4.50 × 10^1^	1.68 × 10^1^	1.67 × 10^2^	8.61 × 10^8^	**1.29 × 10^−1^**
	Std	4.10 × 10^−1^	7.62 × 10^6^	1.19 × 10^7^	8.25 × 10^2^	3.16 × 10^−1^	3.71 × 10^1^	2.36	3.12 × 10^1^	3.28 × 10^8^	7.60 × 10^−2^
	Best	6.36	1.01 × 10^1^	1.87 × 10^6^	1.97 × 10^2^	7.59	9.89	1.26 × 10^1^	8.56 × 10^1^	2.15 × 10^8^	**3.67 × 10^−2^**
F14	Mean	4.50 × 10^−3^	1.64	5.10 × 10^1^	1.05 × 10^2^	7.70 × 10^−5^	1.25 × 10^2^	2.19 × 10^1^	2.46 × 10^1^	8.11 × 10^1^	**2.46 × 10^−6^**
	Std	2.87 × 10^−3^	9.69	1.17 × 10^1^	1.26 × 10^1^	8.55 × 10^−4^	2.23 × 10^1^	3.88	8.16	1.41 × 10^1^	6.63 × 10^−6^
	Best	7.96 × 10^−7^	6.83 × 10^−3^	3.26 × 10^1^	7.72 × 10^1^	5.32 × 10^−12^	6.85 × 10^1^	1.28 × 10^1^	1.50 × 10^1^	5.54 × 10^1^	**1.06 × 10^−14^**
F15	Mean	1.41 × 10^−2^	4.92 × 10^−1^	5.00 × 10^−1^	3.95 × 10^−1^	4.32 × 10^−3^	1.30 × 10^−2^	4.99 × 10^−1^	4.88 × 10^−1^	5.00 × 10^−1^	**3.13 × 10^−3^**
	Std	2.33 × 10^−3^	1.46 × 10^−1^	4.43 × 10^−5^	1.71 × 10^−2^	1.45 × 10^−3^	2.74 × 10^−3^	6.33 × 10^−4^	2.78 × 10^−3^	2.05 × 10^−6^	1.18 × 10^−3^
	Best	9.29 × 10^−3^	1.91 × 10^−2^	5.00 × 10^−1^	3.56 × 10^−1^	3.13 × 10^−3^	6.22 × 10^−3^	4.97 × 10^−1^	4.82 × 10^−1^	5.00 × 10^−1^	**2.58 × 10^−6^**
F16	Mean	1.38 × 10^−10^	8.50	1.07 × 10^3^	7.13 × 10^2^	**7.40 × 10^−18^**	2.48 × 10^−13^	7.18 × 10^1^	1.58 × 10^2^	4.92 × 10^3^	1.87 × 10^−15^
	Std	7.46 × 10^−11^	7.48 × 10^1^	3.89 × 10^2^	9.40 × 10^1^	7.97 × 10^−17^	1.04 × 10^1^	3.26	9.91	1.30 × 10^3^	2.09 × 10^−15^
	Best	2.63 × 10^−11^	4.28 × 10^−1^	5.92 × 10^2^	4.96 × 10^2^	**0.00**	**0.00**	6.25 × 10^1^	1.23 × 10^2^	2.89 × 10^3^	**0.00**

**Table 3 sensors-20-02147-t003:** Running time (seconds) of different algorithms.

Function	GWO	FWA	IPSO	PSO	AGWO	EGWO	BBO	CSA	MFO	FWGWO
F1	0.65625	6.1875	0.828125	0.5625	0.59375	0.53125	10.65625	1.296875	0.953125	1.390625
F2	2.0625	11.5	1.8125	1.828125	2.09375	1.734375	13.89063	4.890625	2.34375	2.46875
F3	0.375	6.75	0.234375	0.15625	0.25	0.171875	9.46875	0.890625	0.453125	0.8125
F4	0.453125	5.5625	0.25	0.1875	0.328125	0.203125	8.671875	0.890625	0.453125	0.671875
F5	0.40625	6.21875	0.21875	0.15625	0.25	0.140625	10.57813	0.75	0.421875	0.671875
F6	1.03125	8.53125	0.8125	0.796875	0.84375	0.71875	9.09375	2.640625	1.0625	2.5
F7	0.609375	6.640625	0.4375	0.375	0.515625	0.375	9.671875	1.296875	0.625	0.90625
F8	0.359375	5.375	0.21875	0.15625	0.234375	0.140625	9.4375	0.71875	0.453125	0.671875
F9	0.421875	6.328125	0.3125	0.25	0.265625	0.1875	8.28125	1.046875	0.453125	0.734375
F10	0.421875	7.171875	0.3125	0.25	0.28125	0.1875	8.3125	0.96875	0.421875	0.703125
F11	0.453125	6.265625	0.328125	0.25	0.34375	0.1875	8.609375	0.90625	0.484375	1.484375
F12	0.953125	7.5625	0.828125	0.71875	1	0.65625	9.09375	2.3125	1.046875	1.28125
F13	1.015625	8.046875	0.796875	0.734375	0.890625	0.703125	9.078125	2.359375	1.046875	1.34375
F14	0.515625	5.875	0.234375	0.21875	0.21875	0.15625	8.5625	0.953125	0.4375	0.625
F15	0.53125	7.25	0.265625	0.1875	0.328125	0.15625	8.125	0.921875	0.421875	0.65625
F16	0.453125	6.109375	0.296875	0.234375	0.28125	0.1875	8.3125	1.015625	0.484375	0.78125

**Table 4 sensors-20-02147-t004:** Wilcoxon’s rank test of FWGWO and other algorithms on 16 benchmark functions.

Function		GWO	FWA	IPSO	PSO	AGWO	EGWO	BBO	CSA	MFO
**F1**	p-value	6.51 × 10^−12^	6.51 × 10^−12^	6.51 × 10^−12^	6.51 × 10^−12^	1.82 × 10^−7^	7.86 × 10^−12^	6.51 × 10^−12^	6.51 × 10^−12^	6.51 × 10^−12^
	h-value	1	1	1	1	1	1	1	1	1
F2	p-value	7.86 × 10^−12^	1.65 × 10^−11^	6.51 × 10^−12^	6.51 × 10^−12^	7.15 × 10^−12^	6.51 × 10^−12^	6.51 × 10^−12^	6.51 × 10^−12^	6.51 × 10^−12^
	h-value	1	1	1	1	1	1	1	1	1
F3	p-value	6.51 × 10^−12^	6.51 × 10^−12^	6.51 × 10^−12^	6.51 × 10^−12^	6.51 × 10^−12^	6.51 × 10^−12^	6.51 × 10^−12^	6.51 × 10^−12^	6.51 × 10^−12^
	h-value	1	1	1	1	1	1	1	1	1
F4	p-value	2.05 × 10^−10^	6.51 × 10^−12^	6.51 × 10^−12^	6.51 × 10^−12^	1.72 × 10^−10^	6.28 × 10^−10^	6.51 × 10^−12^	6.51 × 10^−12^	6.51 × 10^−12^
	h-value	1	1	1	1	1	1	1	1	1
F5	p-value	6.51 × 10^−12^	6.51 × 10^−12^	6.51 × 10^−12^	6.51 × 10^−12^	6.51 × 10^−12^	6.51 × 10^−12^	6.51 × 10^−12^	6.51 × 10^−12^	6.51 × 10^−12^
	h-value	1	1	1	1	1	1	1	1	1
F6	p-value	7.15 × 10^−12^	6.51 × 10^−12^	6.51 × 10^−12^	6.51 × 10^−12^	6.29 × 10^−3^	1.14 × 10^−11^	6.51 × 10^−12^	6.51 × 10^−12^	6.51 × 10^−12^
	h-value	1	1	1	1	1	1	1	1	1
F7	p-value	6.51 × 10^−12^	6.51 × 10^−12^	6.51 × 10^−12^	6.51 × 10^−12^	2.05 × 10^−2^	7.08 × 10^−11^	6.51 × 10^−12^	6.51 × 10^−12^	6.51 × 10^−12^
	h-value	1	1	1	1	1	1	1	1	1
F8	p-value	6.51 × 10^−12^	6.51 × 10^−12^	6.51 × 10^−12^	6.51 × 10^−12^	8.45 × 10^−2^	4.13 × 10^−11^	6.51 × 10^−12^	6.51 × 10^−12^	6.51 × 10^−12^
	h-value	1	1	1	1	0	1	1	1	1
F9	p-value	3.14 × 10^−12^	1.73 × 10^−12^	1.16 × 10^−12^	1.16 × 10^−12^	1.19 × 10^−3^	1.16 × 10^−12^	1.16 × 10^−12^	1.16 × 10^−12^	1.16 × 10^−12^
	h-value	1	1	1	1	1	1	1	1	1
F10	p-value	6.51 × 10^−12^	6.51 × 10^−12^	6.51 × 10^−12^	6.51 × 10^−12^	1.27 × 10^−7^	7.86 × 10^−12^	6.51 × 10^−12^	6.51 × 10^−12^	6.51 × 10^−12^
	h-value	1	1	1	1	1	1	1	1	1
F11	p-value	5.54 × 10^−13^	5.54 × 10^−13^	5.54 × 10^−13^	5.54 × 10^−13^	3.62 × 10^−1^	6.72 × 10^−7^	5.54 × 10^−13^	5.54 × 10^−13^	5.54 × 10^−13^
	h-value	1	1	1	1	0	1	1	1	1
F12	p-value	6.51 × 10^−12^	6.51 × 10^−12^	6.51 × 10^−12^	6.51 × 10^−12^	6.51 × 10^−12^	6.51 × 10^−12^	6.51 × 10^−12^	6.51 × 10^−12^	6.51 × 10^−12^
	h-value	1	1	1	1	1	1	1	1	1
F13	p-value	6.51 × 10^−12^	6.51 × 10^−12^	6.51 × 10^−12^	6.51 × 10^−12^	6.51 × 10^−12^	6.51 × 10^−12^	6.51 × 10^−12^	6.51 × 10^−12^	6.51 × 10^−12^
	h-value	1	1	1	1	1	1	1	1	1
F14	p-value	9.47 × 10^−12^	6.51 × 10^−12^	6.51 × 10^−12^	6.51 × 10^−12^	2.36 × 10^−3^	6.51 × 10^−12^	6.51 × 10^−12^	6.51 × 10^−12^	6.51 × 10^−12^
	h-value	1	1	1	1	1	1	1	1	1
F15	p-value	2.05 × 10^−10^	6.51 × 10^−12^	1.11 × 10^−10^	6.51 × 10^−12^	1.12 × 10^−1^	4.81 × 10^−8^	1.01 × 10^−10^	8.47 × 10^−11^	1.21 × 10^−10^
	h-value	1	1	1	1	0	1	1	1	1
F16	p-value	1.71 × 10^−12^	1.71 × 10^−12^	1.71 × 10^−12^	1.71 × 10^−12^	6.75 × 10^−2^	1.99 × 10^−10^	1.71 × 10^−12^	1.71 × 10^−12^	1.71 × 10^−12^
	h-value	1	1	1	1	0	1	1	1	1

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
