# Peer review of "A Novel Hybrid Algorithm Based on Grey Wolf Optimizer and Fireworks Algorithm"

_sensors, 2020, doi:10.3390/s20072147_

Round 1
Reviewer 1 Report
- The English grammar and expressions need to be improved.
- The author has described research status of GWO, However, some Hybrid Algorithms Based on Grey Wolf Optimizer are ignored; for example, Hybridizing Grey Wolf Optimization with Differential Evolution.
- The author shows that the proposed FWGWO algorithm provides a significant improvement over other algorithms. But time complexity analysis and comparison are not given.
- Running time of different Algorithms are not compared.
Reviewer 2 Report
The paper presents very interesting work and introducing a very sensible improvement to the GWO algorithm. Hence, I recommend it for possible publication.
Reviewer 3 Report
The authors present a hybrid algorithm FWGWO which combines two known optimization algorithms Grey Wolf Optimizer (GWO) and Fireworks Algorithm (FWA) in different stages of the optimization. The algorithm switches between GWO and FWA based on an adaptive balance coefficient with a goal to prevent premature convergence and achieve fast convergence to the global optima. However, I have the following suggestions and concerns about the manuscript.
(i) On line 54-55, authors mention the abbreviations CPSO and SPSO without expanded form and do not provide any references. The reference provided alongside SPSO is a reference to the Fireworks algorithm.
(ii) On line 55, authors mention FWA has a faster convergence speed than some other algorithms, but on line 94, they mention FWA has a slow convergence speed which is contradictory. Authors should mention and elaborate on what cases FWA has slow convergence speed.
(iii) On line 90, authors mention DG and DSTATCOM without providing expanded forms and references.
(iv) On line 48-49, authors should mention and elaborate on what cases GWO converges to local optima rather than just saying some cases.
(v) On line 92-93, authors mention GWO has strong exploitation capability and FWA has high exploration capability. However, on line 95, authors mention that the new algorithm combines the exploitation capability of FWA and exploration capability of GWO. This makes no sense and I believe the authors made a mistake here.
(vi) On line 104, authors should provide a reference for AGWO.
(vii) Between line 133 and 134, there seems to be an extra new line.
(viii) On line 139, vectors r1 and r2 seem to have written in a different font and are bolded which is inconsistent with other vectors.
(ix) All the algorithm pseudocodes (1, 2, and 3) in the manuscript have bad and inconsistent indentations.
(x) The whole manuscript has a lot of inconsistent and missing spaces among the words and symbols. Some examples can be found on line 44, 45, 126, 137, 166, etc.
(xi) The whole manuscript contains a lot of typos and mistakes. For example, line 202, 278, Algorithm 3 (“do” should be “then” in the last if construct), etc.
(xii) The whole manuscript contains inconsistent vertical alignments of variables, constants, and symbols. This is particularly severe on line 188-192.
(xiii) There is no justification provided for equation 16. Why use this equation for balancing? Why not some other equation?
(xiv) The algorithm for FWGWO is not explained adequately. Authors provide very minimal discussion which I find is hardly consistent with Algorithm 3 pseudocode.
(xv) On Algorithm 3, why update the balance coefficient when XÉ‘ is changed? The balance coefficient is not dependent on XÉ‘ according to the equation 16.
(xvi) On Algorithm 3, k is never updated. Also, k and t are never initialized.
(xvii) On Algorithm 3, how are the new search agents selected?
(xviii) On Algorithm 3, why no. of iterations t is only updated when the best fitness is less than XÉ‘? Moreover, this condition is nested in another condition as well.
(xix) On subsection 4.1, no justifications are provided for the experimental settings. Also, instead of just 500 iterations, as the landscapes for the functions are known, the algorithms should be run until the global optimum is found or until they get stuck on local optima.
(xx) Authors should conduct experiments for a number of dimensions for the functions instead of a fixed dimension of 100.
(xxi) Authors provide no justifications for the chosen parameter values presented in Table 1. This makes it hard to tell if a fair comparison is performed.
(xxii) On subsection 4.2, did the authors come up with these test functions? If so, why not use some widely used test functions like Rosenbrock? Also, no. of optima for these functions should be mentioned rather than saying “multimodal functions f9 - f16 have more than two local optima”, which will provide a justification for the claim on line 347 that these are complex spaces.
(xxiii) On subsection 4.3 and Table 2, authors should report minimum fitness solutions as well. Mean and standard deviation could be quite misleading. Reporting minimum fitness solutions would also validate the claims made on line 305-307.
(xxiv) On line 302, authors mention the variances of the experiments of FWGWO are smaller. However, I see larger variances than other algorithms in some cases. What do these cases represent?
(xxv) On figure 3 and 4, are the best score points for just one run or average of 30 runs (which seems unlikely), If for one run, how are these representatives of all the 30 runs? Providing average best-so-far curves could be useful.
(xxvi) For some of the cases in figure 3 and 4, best score curves for FWGWO are not continued until 500 iterations and are stopped abruptly. For example, for F11 function, the curve stops before 300 iterations. Why is that? Also, better color scheme is needed for these figures, some curves are hard to see.
Round 2
Reviewer 1 Report
The paper has been modified based on my suggestion .